# Mapping the Complicated Relationship Between a Temperature Field and Cable Tension by Using Composite Deep Networks and Real Data with Additional Geometric Information

**DOI:** 10.3390/s25175346

**Published:** 2025-08-28

**Authors:** Zixiang Yue, Youliang Ding, Fangfang Geng

**Affiliations:** 1College of Civil Engineering and Architecture, Xinjiang University, Urumqi 830047, China; 007673@xju.edu.cn; 2Key Laboratory of Concrete and Prestressed Concrete Structures of the Ministry of Education, School of Civil Engineering, Southeast University, Nanjing 210096, China; 3School of Architecture Engineering, Nanjing Institute of Technology, Nanjing 211176, China

**Keywords:** cable-stayed bridge, temperature-induced cable tension, deep learning, real data, benchmark model

## Abstract

The abnormal tension change in one cable in a cable-stayed bridge indicates cable damage, so it is necessary to obtain the benchmark of the cable tension. After establishing the regression model of the mapping between the temperature-induced cable tension and the bridge temperature field or other data, the regression value can be used as the benchmark. To improve the regression model, the geometric compatibility and mechanical equilibrium must be jointly considered. Therefore, two data groups, which contain the bridge temperature field and the regression values of the temperature-induced deflection of the main girder, are input into the deep learning neural networks. Time lags exist between the temperature features and the temperature-induced cable tension, but are not significant between the temperature-induced deflection and tension. So one neural network module, which receiving the regression values of the temperature-induced deflection, is composed of Convolutional Neural Networks (CNNs). The other neural network module, which receives the temperature features, is composed of stacked CNN and Long Short-Term Memory (LSTM). Finally, several convolution kernels will integrate the array output from the two modules into one regression value of the temperature-induced cable tension. By combining the input data and the composite neural networks, the *R*^2^ of the regression models of the temperature-induced cable tension is more than 0.95, and the error of the regression values is less than 0.3 kN. In the future, if the nonlinearity at the curve inflection point and the complexity in data distribution could be solved, the stability of the model may be improved.

## 1. Introduction

Cable-stayed bridges are critical nodes in transportation networks, and their optimal operation ensures smooth traffic flow [1,2,3]. In a cable-stayed bridge, compared with the main girder and the cable tower, the section with the cable is slender and has low performance redundancy, which is more likely to get damaged and eventually lead to cable disconnection. Therefore, identification of the damaged cables in a cable-stayed bridge is an eternal pursuit in the field of civil engineering. It is extremely important to automatically and intelligently perceive the state of the bridge in real time, which leads to the bridge health monitoring system; as such, cable damage identification based on monitoring data has become a research hotspot.

A bridge health monitoring system is composed of software content and various sensors installed on the bridge in service. It can continuously measure the temperature, wind, vehicles, and other loads applied on a cable-stayed bridge, as well as the structural responses caused by these loads, such as the tension of the stay cables [4,5]. If the structural performance of the cables deteriorates, the tension of the cables will exhibit abnormal changes, even if the applied load remains constant [6,7,8].

Existing studies utilize ARMA models or deep learning methods to analyze cable tension data through auto-regression, enabling the detection of abnormal time-series information that indicate cable damage.

However, the abnormal parameters of the time-series are mathematically derived and thus do not have clear physical significance, and they only provide qualitative damage indications. This limits quantitative assessment of cable damage. Accurate detection of abnormal tension values is the fundamental for cable damage quantification.

If a high-precision model can be established to correlate the load and response of a bridge in service, the regression value of the response will be output by this model after inputting the measured value of the load, and the regression value of the response can serve as the benchmark of the bridge’s operating status. Deviation between the benchmark and measured values identifies the abnormal structural responses [9,10]. The bridge monitoring system has accumulated extensive real-world operational data, continuously recording the structure’s service performance [11,12]. The input–output model relating structural loads to responses, developed by bridge monitoring big data, accurately represents in-service bridge behavior and has attracted significant research attention [13,14,15]. Determining the loads and the responses introduced by the loads is the first step for modeling.

The state of traffic flow is extremely complex, and the accurate measurement of vehicle loads requires high-end hardware and software support, which is not fully provided by most bridge monitoring systems [16]. Due to the quasi-static characteristic of temperature, it is convenient to accurately measure the structural temperature of a bridge [17]. The measurement of cable tension involves the vibration frequency method [18,19], magnetic flux method [20], pressure ring method [21], etc. Although measurement methods vary, different approaches to cable tension monitoring capture consistent data features for regression modeling, making the choice of measurement method irrelevant for data-driven analysis.

Because the frequency of temperature is far lower than that of wind or vehicle [22], time-frequency multi-scale analysis can effectively extract the cable tension under the combined action of temperature and dead load, that is, the temperature-induced cable tension [23,24]. Since temperature and temperature-induced cable tension data are easier to obtain, mapping models between temperature and temperature-induced cable tension of has become a research focus for cable-stayed bridges [23,24,25].

To establish the mapping model between temperature and temperature-induced cable tension, researchers used linear regression [9,26], machine learning [27,28] and other methods to establish the regression models between the monitoring data of temperature and temperature-induced cable tension. However, the precision of the model established by the existing methods is still insufficient, and the precision of these models is unstable, which has a negative impact on the quantitative perception of the abnormal cable tension based on the benchmark.

Deep learning is generally considered as having stronger nonlinear fitting performance. However, the perception of abnormal cable tension based on deep learning is still focused on time-series analysis, rather than the model of mapping relationship. When using deep learning to establish the mapping model, the first step is to complete the feature engineering, i.e., convert the initial temperature field and structural response of the cable-stayed bridge into the features (feature variables) to express the effective information. At present, there are still limitations in the feature extraction of the temperature field for the modeling of temperature-induced cable tension.

There are more studies on the deep learning models of the relationship between the temperature field and the temperature-induced deflection than the modeling of the temperature-induced cable tension. Existing studies first reduce the dispersed temperature field to limited temperature features using deformation compatibility [29,30], and then employ long-term short-term memory (LSTM) and Convolutional Neural Networks (CNNs) to establish temperature–response correlations. Finally, a precise regression model between the temperature features and the temperature-induced deflection based on the deep learning networks is built [29]. The precision of the model using deep learning is better than those established by linear regression or traditional machine learning.

Deep learning is preferable for establishing the load–response correlations. This necessitates systematic investigation of temperature feature extraction and neural network architectures for modeling temperature-induced cable tension, building on existing research. Two primary methodological approaches emerge as follows:(1)By adopting the concept of transfer learning under the assumption that temperature features affect temperature-induced cable tension in a manner analogous to their influence on temperature-induced deflection, the mapping mode and network architecture developed for modeling temperature-induced deflection can be directly transferred for modeling temperature-induced cable tension.(2)Through mechanical mechanisms, the factors affecting the temperature-induced cable tension of cable-stayed bridges are analyzed, and then the feature variables input into the regression model can be excavated. According to the characteristics of feature variables, the mapping mode and network architecture are formulated to realize the benchmark modeling of temperature-induced cable tension.

Based on real-world monitoring data from an in-service cable-stayed bridge, this study adapts the existing temperature features and neural network architectures (originally developed for modeling the temperature-induced deflection) to establish a mapping model between temperature features and temperature-induced cable tension.

At the same time, this paper will study the mechanical mechanism affecting the temperature-induced cable tension so as to obtain specialized data features for mapping the temperature-induced cable tension, and then establish the deep learning regression model. Then this paper will compare the precision of different methods, and summarize the limitations, to provide support for the improvement in the future.

## 2. Modeling the Temperature-Induced Cable Tension Based on Transfer Learning

In the existing research on how the temperature field of cable-stayed bridges causes the temperature-induced deflection of the main girders, according to the mechanical mechanism, the formulas have been proposed to refined the temperature field into the average temperature of the main girder (*T*_G_), the vertical temperature difference in the main girder (*T*_D_), the tower temperature (*T*_T_), and the cable temperature (*T*_C_) [31,32]. The temperature-induced cable tension and the temperature-induced deflection are both responses caused by the changes in the temperature field, so, based on the idea of transfer learning, the temperature features that affect the temperature-induced cable are also assumed to be *T*_G_, *T*_D_, *T*_T_, and *T*_C_ firstly. Meanwhile, by leveraging the concept of transfer learning, the temperature-induced cable tension will be modeled by the transferred Stack LSTM-CNN from the existing research to verify whether the regression model of the temperature-induced cable tension can be established by the existing method.

### 2.1. Transfer of the Mapping Relationship

The first step to building an effective deep learning model is to construct data features and form a data mapping relationship. Four temperature features input into the deep learning model are *T*_G_, *T*_D_, *T*_T_, and *T*_C_. The temperature-induced cable tension is recorded as *F* which is the output data mapped by the four temperature features. The data input into LSTM should be a time-series but not a data point. Existing research has found that the best modeling effect will be achieved when inputting temperature features during the previous five hours to map the temperature-induced response at the current moment [32,33]. Moreover, the study revealed that resampling the monitoring data at 10 min intervals will achieve an optimal balance between data volume and informational density [29]; therefore, a five-hour period yields 30 data points. So the time-varying mapping relationship from the time point *t* between temperature features and temperature-induced cable tension is shown in Figure 1.

### 2.2. Transfer of the Architecture of the Stack-LSTM-CNN

The time-variable mapping relationship in Figure 1 is used as the supervision learning mode for building the regression model by the deep neural networks. Then, the existing Stack LSTM-CNN architecture [29] is transformed to build the regression model of the temperature-induced cable tension.

Figure 2 shows the architecture of the Stack LSTM-CNN transformed from the existing research [29]. At each time point, the input data matrix, which is constructed in Figure 1, has the size of 4 × 30 (four temperature features and each feature with 30 time-series data points, i.e., feature dimension × time dimension). The first layer of the network has four convolution kernels, including the Convolution Kernel A to Convolution Kernel D as shown in Figure 2. The size of each convolution kernel is 4 × 6. Each 4 × 6 convolution kernel will traverse the 4 × 30 matrix along the time dimension to enhance the information of the temperature features input into the network, and the original matrix size will be convolved to 4 × 25.

After the convolution operation, the 4 × 25 matrix (feature dimension × time dimension) will be input into the second layer with two stacked LSTM hidden layers. For LSTM, the input data is required to be several time-series of different features, and the input data needs to be fed into the corresponding neural network layer at each time step in chronological order. Therefore, LSTM networks require that the dimensions of the input data matrix should be time dimension × feature dimension.

So, the 4 × 25 matrix output by the convolution layer should be transposed to 25 × 4, that is to say, swapping the rows and columns of the matrix. To complete the above matrix transformation, the function “np.transpose ( )” in the NumPy package, based on Python 3.11.7, is used. Next, the 25 × 4 matrix will be input into the second layer with two LSTM hidden layers. Each LSTM hidden layer has 64 cells, thus the second layer will output a 25 × 64 matrix.

Finally, at the last level, the Convolution Kernel E with the size of 25 × 64 is used to convolute the 25 × 64 matrix into a single value, and the linear activation function is used to process this value to obtain the regression value of the temperature-induced response.

To avoid overfitting and to speed up the convergence of model training process, the temperature features are normalized by the Min-Max standardization and input into the network, then the normalized regression value of the temperature-induced response is output from this network. The normalized regression value is de-normalized, and the reference regression value of the temperature-induced response is obtained.

In conclusion, the expression model of the mapping relationship between the temperature features and the temperature-induced cable tension based on the transferred deep learning method is realized. However, the above temperature features are not specifically derived for fitting the temperature-induced cable tension, so the data flow and neural network from transformation may not be suitable for modeling the temperature-induced cable tension. Next, this paper will focus on the special scheme for modeling temperature-induced cable tension.

## 3. Data Features for Modeling Temperature-Induced Cable Tension Based on Physical Interpretation

A cable-stayed bridge represents a complex and highly flexible structure. Without incorporating mechanical principles during data feature construction and neural network architecture design, developing generalizable models becomes challenging.

As a high-order statically indeterminate structure, a cable-stayed bridge permits mechanical analysis at both macroscopic levels (addressing global deformation compatibility and force equilibrium) and rigorous scales (considering nonlinear behaviors and prestress-induced cable tension complexities). While this study primarily adopts data-driven neural network approaches, the structural analysis is focused specifically at the macroscopic level. Consequently, the data set construction in this section incorporates limited physical constraints to facilitate more convenient implementation of the data-driven modeling.

### 3.1. Data Features for Modeling Temperature-Induced Deflection

As shown in Figure 3, under the influence of sunlight and atmospheric temperature, changes in the temperature field of the bridge result in corresponding deformation and displacement of the main girders, cables, and towers. When mapping the temperature features with the temperature-induced deflection, relying on the correlation between “temperature-induced energy change” and “geometric deformation compatibility”, there is no need to disassemble the entire bridge to analyze the mechanical equilibrium, but only to consider the deformation compatibility that the correlation mechanism between the “temperature features (input)” and the “temperature-induced deflection of the main girder (output)”, which can be easily explained by geometrically reasoning.

However, Figure 3 only uses the geometric mechanism and does not break and disassemble the cables into cable tensions. Therefore, based on the four temperature features (*T*_G_, *T*_D_, *T*_T_, and *T*_C_), it is possible to establish a preferable data-driven regression model for mapping cable tensions, or it may not be possible to establish a model with satisfactory precision. Moreover, this transferred method directly applies the existing method from one bridge response to another bridge response; therefore, its physical interpretability is insufficient. This section will first analyze the data features for modeling temperature-induced cable tension based on physical interpretability.

### 3.2. Mixed Data Features for Modeling Temperature-Induced Cable Tension

As shown in Figure 4, to study the temperature-induced cable tension F, the cable system needs to be disassembled, so only considering the coordination between the temperature change and the deformation cannot infer the tension change in each cable, accordingly, the mechanical equilibrium must be considered. Even though the temperature features that cause temperature-induced cable tension can still be intuitively defined as *T*_G_, *T*_D_, *T*_T_, and *T*_C_, the complexly dynamic relationship between temperature features, geometric deformation, and mechanical equilibrium obviously leads to modeling difficulties.

The input data with sufficient information of physical interpretation is the key point to establish a high-precision digital regression model. It is not enough to model the temperature-induced cable tension only by inputting four temperature features *T*_G_, *T*_D_, *T*_T,_ and *T*_C_. As shown in Figure 5, focusing on the load conditions and geometry of the main girder, the gravity of the main girder is one type of “directional load” that does not change in value and direction, and the sum of the vertical components of each cable should be equal to the gravity of the main girder. In addition to the cable tension value *F*, the vertical component is determined by the angle between the cable and the horizontal direction, because the head end of the cable is fixed to the main girder and its tail end always intersects at one point with the tail end of the other cable, so the change in this angle depends on the change in the deflection of the main girder and the change in the length of the cable. The cable is a tension spring, so the change in cable tension actually represents the change in cable length.

Therefore, the changes in the deflection of the main girder and the cable tension caused by the change in temperature field affect the mechanical equilibrium, and the temperature features correspond to the two unknown variables of deflection and cable tension, so it is naturally impossible to establish a high-precision deep learning model which outputs the regression value as the benchmark value of the cable tension.

On the basis of inputting four kinds of temperature features, this chapter will use the high-precision regression value of temperature-induced deflection obtained by the deep learning model as the supplemental information of the geometry of the main girder to enhance the information abundance with the mixed input data of “temperature features + regression value of deflection the main girder” to obtain a more accurate regression model of temperature-induced cable tension. Next, the bridge health monitoring system and data set used in this paper are first introduced. Then, based on the information of the data set, this paper explains how to input the regression value of temperature-induced deflection to enhance the input information.

## 4. Data Preparation and Mapping the Relationship Between Cable Tension and Temperature and Deflection

The study is driven by the real data obtained by the monitoring system on an in-service cable-stayed bridge. Firstly, the bridge health monitoring system will be introduced.

### 4.1. Bridge Health Monitoring System

The data in this paper come from a large cable-stayed bridge in service, an important part of the cross-sea bridges in southern China. As shown in Figure 6, this bridge is a two-tower cable-stayed bridge, the tower is made of concrete, and the main girder consists of a flat steel box. The health monitoring system of this bridge can monitor the temperature of the main girder, the temperature of the cable tower and the cable temperature, as well as the deflection of the main girder and the cable tension of some cables [34].

As shown in Figure 7a, the bridge has the total length of 1150 m consisting the spans of 110 m + 236 m + 458 m + 236 m + 110 m. Due to the symmetry of cable-stayed bridges, the vertical deflection of the main girder and the cable tension only on one side are analyzed. Meanwhile, since the 110 m outermost side span is structurally supported by an auxiliary pier without being integrated with the cable-staying system, it has been omitted from the analytical model.

As shown in Figure 7a, five deflection sensors (differential pressure transmitter) are installed on the west half of the main girder, from the mid of the main span to the mid of the side span, denoted as D1~D5. On the south side of the western half of the bridge, the tension sensors C1 and C2 based on the vibration frequency method are installed on the cables linking the tower with the mid of the main span of the main girder and the mid of the side span of the main girder.

All temperature sensors in the bridge health monitoring system are fiber Bragg grating (FBG) sensors. Because the cable temperature has little difference longitudinally and the cable section is not complex, as shown in Figure 7a, the temperature sensor CT1, on the cable in the mid of the side span, is sufficient to represent the cable temperature. The monitoring data of the temperature field of the main girder also has little difference longitudinally, so only the temperature measured by the sensors installed on section 1-1 of the main girder of the mid of the main span is used. Section 1-1 is shown in Figure 7b; the 12 temperature sensors, which are installed on the main girder, are recorded as T1~T12.

The temperature of the tower has little difference vertically, so it is sufficient to only take the section of the tower at a certain height for monitoring. As shown in Figure 8a,b, a total of four concrete temperature sensors are installed on section 2-2 of the west bridge tower, which are recorded as T13~T16.

Then, the temperature features and temperature-induced responses required for data-driven research will be introduced.

### 4.2. Temperature Features and Temperature-Induced Responses

The bridge monitoring big data should be refined into several data features expressing the essence of bridge mechanics. In the existing research on how the temperature field of cable-stayed bridges causes the temperature-induced deflection of the main girders, according to the mechanical mechanism [31], the formulas have been proposed to refine the temperature field into the average temperature of the main girder (*T*_G_), the vertical temperature difference in the main girder (*T*_D_), the tower temperature (*T*_T_), and the cable temperature (*T*_C_) [31,32]. The temperature-induced cable tension and the temperature-induced deflection are both responses caused by the changes in the temperature field, so the temperature features that affect the temperature-induced cable are also assumed to be *T*_G_, *T*_D_, *T*_T,_ and *T*_C_. Additionally, the temperature-induced cable tensions should be extracted, and the temperature-induced deflections provided as supplementary information should also be extracted.

For extracting the temperature-induced response caused by dead load and temperatures from the original signal, in the existing research, the time-frequency analysis based on Daubechies Wavelet multi-scale decomposition can effectively extract the temperature-induced information in the structural response [29,30,35]. Here, the study also uses the Wavelet method to extract the temperature-induced responses in the original signals.

The temperature-induced response is triggered by temperature, so the frequency domain information of the temperature-induced response should be close to the frequency domain information of temperature data.

So, the correlation coefficients between the frequency spectra of the temperature-induced responses (extracted using different wavelet parameters) and the frequency spectrum of the temperature data are analyzed, and the extracted result with the highest frequency correlation coefficient are used as the determined temperature-induced deflection and tension.

Figure 9 shows the temperature-induced deflection and the temperature-induced cable tension obtained by the Wavelet time-frequency analysis for one day.

Next, the data sets used in this paper will be described.

### 4.3. Data Set

The data of the temperature field, the deflection of the main girder, and the cable tension of this in-service cable-stayed bridge from January to December 2020 were obtained through the health monitoring system. To align the time points of the multi-source data sets and control the data scale not only expressing the characteristics of temperature change, but also not affecting the modeling efficiency [29], all data sets are resampled as 10 min intervals between each data point [29], and then the NaN values were removed to form the uninterrupted time-series. *T*_G_ is the average value of the temperature obtained by sensors T1–T12, *T*_D_ is the difference between the average temperature of the data obtained from sensors T1 to T8 and the average temperature the data obtained from sensors T9 to T12, *T*_T_ is the average value of the data obtained from sensors T13 to T16, and *T*_C_ is the data obtained from of sensor CT1. The temperature-induced deflections *D*_1_, *D*_2_, *D*_3_, *D*_4_, and *D*_5_ are constructed by the deflection data collected from sensors D1, D2, D3, D4, and D5, respectively, and the temperature-induced cable tensions *F*_1_ and *F*_2_ are constructed by the cable tension data collected from sensors C1 and C2, respectively.

The temperature features *T*_G_, *T*_D_, *T*_T,_ and *T*_C_ are shown in Figure 10. Each temperature variable has 47,520 data points. As the data obtained by the monitoring system is quite complete, the first 60% data for each data set, with 28,512 data points, is taken as the training set, the middle 20%, with 9504 data points, as the verification set, and the last 20%, with 9504 data points, as the test set.

The temperature-induced deflection features *D*_1_, *D*_2_, *D*_3_, *D*_4,_ and *D*_5_ are shown in Figure 11. Each deflection feature has 47,520 data points. The first 60% data for each data set, with 28,512 data points, is taken as the training set, the middle 20%, with 9504 data points, as the verification set, and the last 20%, with 9504 data points, as the test set.

The temperature-induced cable tension features *F*_1_ and *F*_2_ are shown in Figure 12. There are 47,520 data points for each tension feature. The first 60% data for each data set, with 28,512 data points, is taken as the training set, the middle 20%, with 9504 data points, as the verification set, and the last 20%, with 9504 data points, as the test set.

### 4.4. The Mapping Relationship Between Temperature, Deflection, and Cable Tension

The input data of the regression model is the mixed information of temperature and temperature-induced deflection to map the temperature-induced cable tension. The regression value of temperature-induced cable tension serves as the benchmark for the normal service state. Therefore, the supplemental main girder geometry input into the regression model should reflect the deflection under normal conditions. Thus, the temperature-induced deflection input should be the regression value (benchmark) from the deep learning model, not the measured deflection.

Similarly to the temperature-induced deflection as shown in Figure 13, the temperature-induced cable tension *F*_1_ also exhibits a significant time lag relative to temperature. Since deflection (*D*_1_) and tension (*F*_1_) are both structural responses, they should theoretically have minimal time lag between them. This is confirmed in Figure 14, where only a slight delay is observed between *D*_1_ and *F*_1_.

Therefore, the regression data of the temperature-induced deflection and the temperature features should be taken as two subsets, respectively, to be input into the deep learning model, so as to avoid the disharmony of the difference in the time lags, which will lead to the inadequate modeling precision and even the failure of the convergence during the training phase of the model. Considering this factor, the description of the two subsets is as follows.

As shown in Figure 15, the existing four temperature features and the time-series of 30 data points for each feature constitute the input data Group A. For the bridge used in this paper, considering the symmetry of the main girder, the regression values of temperature-induced deflection of the five deflection measurement points (*D*_1_~*D*_5_) constitute the Input Data Group B to supplement the geometry information of the main girder. The Input Data Group A and the Input Data Group B constitute the mixed input data of the input neural network.

The data of Group A and Group B need to be integrated to one value to obtain the regression value of the temperature-induced cable tension. Therefore, the scale of the two input data groups should remain the same scale in the time dimension, that is, each time-series of the regression value of the temperature-induced deflection of the Group B also contains 30 data points.

For the two data groups, two independent neural network modules are required for processing the two data groups separately, and another independent neural network module is used for data integration to output the regression value of the cable tension. Thus, this chapter will develop a deep learning model with composite neural networks.

## 5. Modeling Temperature-Induced Cable Tension Based on Composite Neural Networks

To fuse the data Group A and data Group B, it is necessary to formulate the neural network Module A and neural network Module B according to the characteristics of the two data groups, and finally integrate the output values of the two modules through neural network Module C to obtain the regression value *F*’ of cable tension *F*.

In the previous discussion, the distinct physical meanings of data Group A and data Group B, as well as their differing temporal lag correlation characteristics with output data, have been explored. Consequently, Module A and Module B are constructed using different types and levels of neural networks to accommodate the temporal lag disparities between the data groups. Meanwhile, Modules A, B, and C are all designed to process data correctly along the temporal direction. Following this rationale, the neural network architecture will gain enhanced physical interpretability while maintaining the robust structure and temporal coherence when handling complex, high-dimensional input data.

Figure 16 shows the architecture of the neural networks composed by Module A, Module B, and Module C. The detailed description of neural network architectures and data flow is provided in the following subsections.

### 5.1. Neural Network Module A

First, the Neural Network Module A is introduced. The network Module A is shown in Figure 17. Module A receives the input data Group A, and data Group A is just the four temperature features *T*_G_, *T*_D_, *T*_T,_ and *T*_C_ in Section 3, so network Module A is basically the same as the Stack CNN-LSTM in Section 2.2 but without Convolution Kernel E, and the output information of the Network Module A is a 25 × 64 data matrix (time dimension × feature dimension). To facilitate the subsequent Network Module C to process data along the time dimension, the output data matrix is transposed to 64 × 25 (feature dimension × time dimension), that is, the Output Group A as shown in Figure 17.

The 64 × 25 matrix of Output Group A is marked as ***M***_A_, and ***M***_A_ is expressed as follows:(1)$$M_{A}=\left[\begin{array}{cccc} {h^{\prime }}_{1(1)} & \ldots & {h^{\prime }}_{1(24)} & {h^{\prime }}_{1(25)}\\ {h^{\prime }}_{2(1)} & \ldots & {h^{\prime }}_{2(24)} & {h^{\prime }}_{2(25)}\\ \ldots & \ldots & \ldots & \ldots \\ {h^{\prime }}_{64(1)} & \ldots & {h^{\prime }}_{64(24)} & {h^{\prime }}_{64(25)} \end{array}\right]$$
where taking *h*′_1(24)_ as an example, this value represents the matrix coefficient of column 24 in row 1.

### 5.2. Neural Network Module B

There is almost no time lag between the temperature-induced deflection and the temperature-induced cable tension, so there is no temporal nonlinearity between them so there is no need to use LSTM to process the Input Data Group B. Therefore, the Network Module B only uses multiple convolution kernels to form the CNN to process the Input Data Group B. Network Module B and its output matrix ***M***_B_ are shown in Figure 18. The Network Module B is divided into two parts, one of which is named the Feature Modulation and the other is named the Temporal Modulation, and then the two matrices processed by the two processes are superimposed as ***M***_B_ which is Output Data Group B. Next, the Network Module B will be described in detail.

The significance of the Feature Modulation is to enhance the local information of the input feature. The Feature Modulation uses five convolution kernels (Kernel A~Kernel E, in_channels = 5, out_channels = 1), respectively, to traverse the Input Data Group B, and then obtains the 5 × 25 matrix ***M***_B1_. Because of the low depth of the network of Feature Modulation, ***M***_B1_ can be concurrently used as the residual connection information to prevent over fitting.

The purpose of the Temporal Modulation is to perform multiple time-shifting convolutions on the Input Data Group B through several convolution kernels to enhance the temporal characteristics of the data set. The Temporal Modulation uses five 5 × 2 convolution cores (Kernel F~Kernel J, in_channels = 5, out_channels = 5). In this calculation process, each convolution kernel forms a layer of the network, and the Input Data Group B is processed successively by Kernel F to Kernel J, and finally the 5 × 25 matrix ***M***_B2_ is output.

To enhance the robustness of the network, ***M***_B1_ and ***M***_B2_ are processed by the ReLU activation function, and then the matrix ***M***_B1_ and ***M***_B2_ are added to obtain the 5 × 25 matrix ***M***_B_, which is the Output Group B. The ReLU activation function has been embedded in Pytorch 2.3.0, as shown in Equation (2):(2)$$ReLU(x)\;=\;R(x)=\left\{\begin{array}{c} x,\;\mathrm{if}\;x>0\\ 0,\;\mathrm{if}\;x\leq 0 \end{array}\right.$$

As shown in Figure 19, to illustrate the data flow in the Network Module B, the data flow is introduced by taking *y*′_30_ as the regression value at the 30th time point as an example to introduce this data flow. *d*_1(30)_ is the normalized regression value of *D*′_1_ the deflection of the main girder at the middle of the main span at the 30th time point; therefore, the normalized *D*′_1_ to *D*′_5_ form the normalized Input Data Group B is as shown in Equation (3):(3)$$\left[\begin{array}{ccccc} d_{1(1)} & d_{1(2)} & \ldots & d_{1(29)} & d_{1(30)}\\ d_{2(1)} & d_{2(2)} & \ldots & d_{2(29)} & d_{2(30)}\\ d_{3(1)} & d_{3(2)} & \ldots & d_{3(29)} & d_{3(30)}\\ d_{4(1)} & d_{4(2)} & \ldots & d_{4(29)} & d_{4(30)}\\ d_{5(1)} & d_{5(2)} & \ldots & d_{5(29)} & d_{5(30)} \end{array}\right]$$

As shown in Figure 19 the normalized Input Data Group B is first brought into the neural network of the Feature Modulation, which is the same as the Convolution Layer 1 of the Network Module A. One convolution will convert the 5 × 6 data in the receptive field of the convolution kernel into one single value (out_channels = 1), and each convolution kernel will traverse along the time dimension for 25 times to process the 5 × 30 input data into a 1 × 25 vector. If the vector obtained by Kernel A is [*x*′_A(1)_, *x*′_A(2)_, … *x*′_A(25)_], the matrix ***M***_B1_ is composed of five vectors obtained by the five convolution Kernels A~E, respectively, traversing five times, as follows:(4)$$M_{B1}=\left[\begin{array}{ccccc} x\prime_{A(1)} & x\prime_{A(2)} & \ldots & x\prime_{A(24)} & x\prime_{A(25)}\\ x\prime_{B(1)} & x\prime_{B(2)} & \ldots & x\prime_{B(24)} & x\prime_{B(25)}\\ x\prime_{C(1)} & x\prime_{C(2)} & \ldots & x\prime_{C(24)} & x\prime_{C(25)}\\ x\prime_{D(1)} & x\prime_{D(2)} & \ldots & x\prime_{D(24)} & x\prime_{D(25)}\\ x\prime_{E(1)} & x\prime_{E(2)} & \ldots & x\prime_{E(24)} & x\prime_{E(25)} \end{array}\right]$$

As shown in Figure 19, the Input Data Group B is brought into the Feature Modulation which is the network referring to the TCN (Temporal Convolutional Networks) [36] with five layers (each layer is a 5 × 2 convolution kernel with out_channels = 5, i.e., Kernel F~Kernel J) is used to successively traverse the input data. After passing through one layer of this network, the time dimension of the data matrix is reduced once, and the last layer Kernel J outputs ***M***_B2_ as follows:(5)$$M_{B2}=\left[\begin{array}{ccccc} x\prime_{J1(1)} & x\prime_{J1(2)} & \ldots & x\prime_{J1(24)} & x\prime_{J1(25)}\\ x\prime_{J2(1)} & x\prime_{J2(2)} & \ldots & x\prime_{J2(24)} & x\prime_{J2(25)}\\ x\prime_{J3(1)} & x\prime_{J3(2)} & \ldots & x\prime_{J3(24)} & x\prime_{J3(25)}\\ x\prime_{J4(1)} & x\prime_{J4(2)} & \ldots & x\prime_{J4(24)} & x\prime_{J4(25)}\\ x\prime_{J5(1)} & x\prime_{J5(2)} & \ldots & x\prime_{J5(24)} & x\prime_{J5(25)} \end{array}\right]$$

As shown in Figure 19, ***M***_B1_ and ***M***_B1_ are processed by ReLU activation function and added to obtain the output data matrix ***M***_B_ as follows:(6)$$M_{B}=\;R(M_{B2})+R(M_{B2})\;=\;\left[\begin{array}{ccccc} d\prime_{1(1)} & d\prime_{1(2)} & \ldots & d\prime_{1(24)} & d\prime_{1(25)}\\ d\prime_{2(1)} & d\prime_{2(2)} & \ldots & d\prime_{2(24)} & d\prime_{2(25)}\\ d\prime_{3(1)} & d\prime_{3(2)} & \ldots & d\prime_{3(24)} & d\prime_{3(25)}\\ d\prime_{4(1)} & d\prime_{4(2)} & \ldots & d\prime_{4(24)} & d\prime_{4(25)}\\ d\prime_{5(1)} & d\prime_{5(2)} & \ldots & d\prime_{5(24)} & d\prime_{5(25)} \end{array}\right]$$
where *R*( ) stands for ReLU activation function.

Thus, the introduction of Network Module B has been completed and the Output Group B ***M***_B_ has been obtained. Next, the Network Module C is designed to integrate ***M***_A_ and ***M***_A_ to obtain the regression value *y*′_30_.

### 5.3. Neural Network Module C

The Neural Network Module C is used to integrate the output data ***M***_A_ from the Network Module A and the output data ***M***_B_ from the Network Module B into the single regression value before inverse normalization. The network architecture of the Neural Network Module C is shown in Figure 20.

After being processed by different neural network modules, the extreme values and data distributions of ***M***_A_ and ***M***_B_ differ greatly. To prevent this problem from leading to network training failure, ***M***_A_ and ***M***_B_ need to be, respectively, normalized before be put into the Neural Network Module C. Therefore, as shown in Figure 20, we first use the batch normalization function (the BN layer in Pytorch platform) [37] to normalize ***M***_A_ and ***M***_B_, and then splice the normalized ***M***_A_ (size 5 × 25) and the normalized ***M***_B_ (size 5 × 25) along the longitudinal (feature dimension) to obtain the input matrix ***M***_C_, so the size of ***M***_C_ is 10 × 25.

The Neural Network Module C with five convolution kernel uses the idea of TCN, taking each convolution kernel as a layer of the network, and each kernel successively traverses the input data along the time dimension, and finally changes the MC with “time-feature” dual dimensions into a single regression value through several convolutions.

Firstly, Kernel K compress the two dimensions of ***M***_C_. The Kernel K (size 69 × 2, out_channels = 5) is used to traverse the ***M***_C_ along the time dimension. The feature dimension is compressed from 10 to 5, and the time dimension is reduced to 24. Thus the Kernel K output the matrix ***M***_K_ with size 5 × 24.

The receptive field of the convolution kernel with a lateral scale of two is relatively small. The smaller lateral receptive field will require a lot of convolution layers to reduce the time dimension of information. If there are too many convolution layers or the lateral size of convolution kernel is too large, the network will have the tendency of overfitting. So the Convolution Kernel L starts to use dilated convolution, adding holes between convolution elements to increasing the receptive field without increasing the size of convolution kernel. At the same time, to speed up the compression of the feature dimension, the out_channels of the convolution kernel are decreased in turn. Kernel L has the size of 5 × 2 and out_channels = 4, and the dilation number of 4 (there are three holes between the two convolution elements). Receiving the matrix from the upper layer, the Kernel L will output a 4 × 20 matrix ***M***_L_.

Kernel M has the size of 4 × 2 and out_channels = 3, and the dilation number of 6 (there are five holes between the two convolution elements). Receiving the matrix from the upper layer, the Kernel M will output a 3 × 14 matrix ***M***_M_.

Kernel N has the size of 3 × 2 and out_channels = 2, and the dilation number of 6 (there are five holes between the two convolution elements). Receiving the matrix from the upper layer, the Kernel N will output a 2 × 8 matrix ***M***_N_.

Kernel Z is responsible for converting the output matrix ***M***_N_ to the regression value *y*′_0_ at the 30th time point. Therefore, the size of Kernel Z is 2 × 8, and out_channels = 1. Kernel Z will output *y*′_30_ which is the normalized value; after inverse normalization, the normalized regression value *y*′_30_ will be formed as the regression value *F*′ of cable tension at the 30th time point.

## 6. Regression Models of Temperature-Induced Cable Tension

The transferred Stack-LSTM-CNN and the composite networks are established by the PyTorch platform. The CPU of the computer is Core-i7-14650HX with 5.2 GHz, and the memory of the computer is 32 GB. The training set and the verification set are normalized before input into the deep neural networks. Mean Squared Error (*MSE*) is calculated by Equation (7) as the loss function during the training process.(7)$$MSE=\frac{1}{N}\sum_{n=1}^{N}{\left(y_{n}-y_{n}^{\prime }\right)}^{2}$$
where *y_n_* is the measured value; *y*′*_n_* is the regression value; and *N* is the total number of data points in the data set.

Training a qualified deep learning model requires further utilization of the test set to check whether the model has generalization performance. The Mean Squared Error (*MSE*) can be used to evaluate the volatility of the output values of the trained model. Additionally, the Mean Absolute Error (*MAE*) is used to assess the precision of the model. The *MAE* is calculated as shown in Equation (8):(8)$$MAE=\frac{1}{n}\sum_{n=1}^{N}\left|y_{n}-y_{n}^{\prime }\right|$$

Meanwhile, the goodness of fit *R*^2^ is used to assess if the regression values could explain the true values. *R*^2^ is calculated as Equation (9):(9)$$R^{2}=1-\frac{\sum_{n=1}^{N}{\left(y_{n}^{\prime }-y_{n}\right)}^{2}}{\sum_{n=1}^{N}{\left(y_{n}-\bar{y}\right)}^{2}}$$

### 6.1. Regression Model of Temperature-Induced Cable Tension by Transferred Stack-LSTM-CNN

Build two regression models for temperature-induced cable tension *F*_1_~*F*_2_ using the transferred neural network architecture, through the training with 1000 epochs. Normalize the training and validation sets, using *MSE* which is calculated by Equation (7) as the loss function during the training process. The normalized loss curves of the training process for the two models are shown in Figure 21. After 2500 epochs of training, the loss curves of the deflection *F*_1_ and *F*_2_ models showed the error convergence trend in the training set, but the error curve of the validation set was significantly different from the curve of the training set. After training, the test set was used for testing and evaluation.

Record the regression values of *F*_1_~*F*_2_ as *F*′_1_~*F*′_2_. After inputting the test set data of cable tension *F*_1_ and *F*_2_ into the trained models, the *MSE*, *MAE*, and *R*^2^ calculated by *F*_1_~*F*_2_ and *F*′_1_~*F*′_2_ of the two models are as shown in Table 1. As shown in Table 1, the *MSE* and *MAE* of the regression models for *F*_1_ and *F*_2_ are relatively small, with the *R*^2^ exceeding 0.8 but below 0.9, indicating that the fitness between the measured value and the regression value maybe unsatisfactory.

Figure 22 and Figure 23 compare the actual value curves of *F*_1_ and *F*_2_ and the regression value curves of *F*′_1_ and *F*′_2_, and it can be observed that the actual value and the regression value do not fit significantly at the peaks and valleys. As shown in Figure 24 and Figure 25, the scatter diagram of the correlation between the actual values and the regression values shows that the relationship between the actual values and the regression values shows a certain discreteness, regardless of *F*_1_ or *F*_2_, indicating that the Stack-LSTM-CNN is still not enough to fit the complex correlation between the temperature field and the temperature-induced cable tension of cable-stayed bridges. Next, the model based on the composite neural networks will be tested.

### 6.2. Regression Model of Temperature-Induced Cable Tension by Composite Neural Networks

Based on the above improved composite neural networks and mixed input data, the network training and testing are carried out for *F*_1_ and *F*_2_, respectively, and the training phase and test phase are not changed. The input data set consists of temperature features mixed with the benchmark values of the deflection of main girder, so the regression models of deflection *D*_1_~*D*_5_ are established. The objective of this paper is to establish a regression model for cable tension. Therefore, the regression model for deflection will not be elaborated further [29].

The convergence processes of the five deflection regression models during training are shown in Figure 26, with good convergence effect. After the training phase, the test set will be used for evaluating whether the five models have generalization performance and accuracy.

After inputting the test sets of *D*_1_~*D*_5_ into the trained model, *MSE*, *MAE*, and *R*^2^ of the five models are as shown in Table 2. The *MSE* and *MAE* of the regression results of *D*_1_~*D*_5_ are all small, showing the high precision and stability of five deep learning models. More critically, the *R*^2^ value should be prioritized for examination, as its magnitude directly indicates the explanation of the regression data to the real data.

As shown in Table 2, the *R*^2^ of five models are all more than 0.95, indicating that the five models can accurately express the thermal benchmark of the temperature-induced deflection.

During the training phases of two composite neural networks of *F*_1_ and *F*_2_, the loss curves of the training sets and the verification sets are shown in Figure 27. After the training, the loss curves of the training sets and the verification sets of the two models both show the error convergence trend. After the training, the two models will be tested.

After inputting the test sets of cable tension *F*_1_ and *F*_2_ into the trained models by the composite neural networks, the *MSE*, *MAE*, and *R*^2^ of the two models are as shown in Table 3. As shown in Table 3, the *MSE* and *MAE* of the regression models by the composite neural networks for *F*_1_ and *F*_2_ are relatively small, with the *R*^2^ exceeding 0.9, indicating that the excellent fitness of the models build by the composite neural networks.

As shown in Figure 28 and Figure 29, when fitting the cable tension *F*_1_ by the deep learning regression model based on mixed data and composite neural network, the correlation between the actual value and the regression value is good, but some regions realize a certain degree of discreteness. The actual value curve and the regression value curve fit well, but the error at the peak is relatively small, while the error at the valley is relatively large.

As shown in Figure 30 and Figure 31, when fitting the cable tension *F*_2_ by the deep learning regression model based on mixed data and composite neural network, the correlation between the actual value and the regression value is good, but some regions realize a certain degree of discreteness. The actual value curve and the regression value curve fit well, but the error at the peak is also relatively small, while the error at the valley is also relatively large.

To summarize, based on the mixed input data and composite neural network, the precision of fitting the temperature-induced cable tension is greatly improved, but the precision difference between the peak and valley values is large, which will significantly affect the stability of the output value of the regression model, which should be analyzed.

### 6.3. Error Characteristics in Fitting Temperature-Induced Cable Tensions

The absolute error (*AE*) of the regression models of *F*_1_ and *F*_2_ is analyzed. The calculation equation of *AE* is shown as follows:(10)$$AE=y_{n}-y_{n}^{\prime }$$

As shown in Figure 32, the absolute error curves of *F*_1_ and *F*′_1_ are drawn, and the histogram is drawn by the absolute error data as shown in Figure 33. It can be observed that the absolute error is not the zero mean value but a sine wave, like the change trend similar to the temperature-induced response. At the same time, the shape of the histogram is not symmetrical with the average value, so it is not suitable to weaken the error by sliding average.

As shown in Figure 34, the absolute error curves of *F*_2_ and *F*′_2_ are drawn, and the histogram is drawn by the absolute error data as shown in Figure 35. For *F*_2_, the same phenomenon as *F*_1_ has been observed.

The non-zero-Mean Absolute Error is clearly linked to inconsistencies in peak and valley error values, though the exact cause is complex and will be analyzed later. First, we examine the reasons behind the sinusoidal variation in absolute error.

#### 6.3.1. The Analysis of the Sine Wave Shape on the Absolute Error

As shown in Figure 36, the true value, regression value, and absolute error near one peak of *F*_1_ are plotted. It can be seen that the closer to the peak, the greater the error. The reason is that the closer it is to the peak, the stronger the nonlinearity of the temperature response and temperature feature, and the more difficult it is to express the nonlinear relationship by the regression model, so the error near the peak is greater.

#### 6.3.2. Analysis of the Asymmetry on the Error Distribution

As shown in Figure 37, the temperature-induced cable tension curve is divided into two parts by the middle point of the curve between the adjacent peak point and valley point. The part which contains the peak value is named the “upper part”, and the part which contains the valley value is named the “lower part”. As shown in Figure 38, there exists the phase difference of half a cycle between the two curves of *F*_1_ and *F*_2_. For *F*_1_ and *F*_2_, the peak value of *F*_1_ is actually at the same time as the valley value of *F*_2_, so the upper part of *F*_1_ should be similar to the lower part of *F*_2_, and the lower part of *F*_1_ should be similar to the upper part of *F*_2_.

The histograms of the upper and lower parts of *F*_1_ are shown in Figure 39a,b, respectively. The upper part exhibits a symmetrical distribution, while the lower part is asymmetrical. When training a deep learning regression model with *MSE* loss, the objective is to minimize global error. However, due to the significant difference in distributions between the two parts, the model can only prioritize fitting one distribution closely—inevitably compromising accuracy in the other. Intriguingly, the model tends to favor fitting the upper (larger-value) portion, though the underlying reason remains elusive.

The histograms of upper part and lower part of *F*_2_ are shown in Figure 40a,b. It is also found that the histogram of the upper part shows an asymmetrical distribution, and the histogram of the lower part shows a symmetrical distribution, which is the opposite of the situation of *F*_1_. This is because there exists the phase difference of half a cycle between the two curves of *F*_1_ and *F*_2_, as shown in Figure 38. As with *F*_1_, the regression model of *F*_2_ also tends to be more precise in the upper part, indicating that the neural network tends to more accurately fit the larger values rather than a certain distribution, but the reason for this phenomenon is elusive.

The above analysis shows that the modeling method in this paper has made progress compared with the existing method, but there are still two problems: the unstable output precision and the asymmetric numerical distribution of error. Follow-up research should pay attention to solve these problems.

### 6.4. Discussion

This paper develops a composite neural network regression model and the corresponding mixed input data for mapping temperature-induced cable tension, whose output serves as the benchmark for cable condition monitoring. Therefore, the higher the precision of the regression value output by this model, the more helpful it is to use the benchmark to detect the damaged state as soon as possible. However, the model still has two problems: Firstly, reduced precision at peaks/valleys due to stronger nonlinearity; and secondly, biased fitting favoring larger values over smaller ones.

Regarding nonlinear errors, future work should incorporate more detailed physical mechanisms (particularly the nonlinear second-order effects and cable prestressing in cable-stayed bridges) into data construction, model design, and error parameter formulation. This will require subsequent studies involving complex mechanical analysis and finite element model updating.

In summary, to solve the disadvantages of the model, further exploration should be conducted on the following perspectives:(1)Study the correlation between temperature field, temperature-induced deflection of the main girder, temperature-induced deformation of the tower, and temperature-induced cable tension to ensure that the input information has sufficient and improved mechanical interpretability.(2)Analyze the temporal characteristics of the bridge temperature field and the temperature-induced responses considering the nonlinear second-order effects and cable prestressing, and determine how to improve the neural network architecture and the modeling method.(3)Improve the form of loss function in the training process of the neural network. Using *MSE* as the loss function makes the global error smaller but *MSE* cannot characterize the local error, so it is worth thinking about how to pay attention to the local error based on the high-order nonlinear analysis of cable-stayed bridges.

## 7. Conclusions

The temperature-induced cable tension of a cable-stayed bridge represents the mechanical performance of the cable. To obtain the benchmark value of the temperature-induced cable, this paper investigates the deep learning-based regression model of the mapping relationship between the temperature-induced cable tension and the bridge temperature field mixed with the temperature-induced responses. Through the research, this paper finds the following:(1)The input data, with physical interpretability and sufficient information, is necessary for ensuring good precision of the regression model. Considering geometric compatibility and geometric compatibility, the input data to establish the temperature-induced deflection regression model is divided into two data groups. One of the data groups consists of four temperature features: the average temperature of the main girder, the vertical temperature difference in the main girder, tower temperature, and cable temperature; the other data group is the regression values of the temperature-induced deflection of the main girder.(2)Deep learning neural networks have strong performance on nonlinear fitting and can thus process the input data with high dimensions, maximizing the information potential of big data. Two neural network modules are developed for processing the four temperature features and the temperature-induced deflection, and a third module is developed to integrate the output data sets from the first two modules to output the regression value of the temperature-induced cable tension. The deep learning regression model composed of three neural network modules has better precision, with R2 over 0.95 and an average error less than 0.3 kN.(3)The improved deep neural network, which is used for regressing the temperature-induced cable tension, still has problems of output precision that is not stable enough and the tendency to more precisely fit larger values, because there are still imperfections in dealing with the local nonlinearity and the complex data distribution patterns, indicating that the method in this paper still has the potential to be improved.

## Figures and Tables

**Figure 1 sensors-25-05346-f001:**
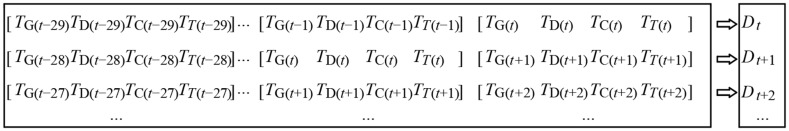
The time-varying mapping relationships between *T*_G_ and *T*_D_ and *T*_T_ and *T*_C_ and *F*.

**Figure 2 sensors-25-05346-f002:**
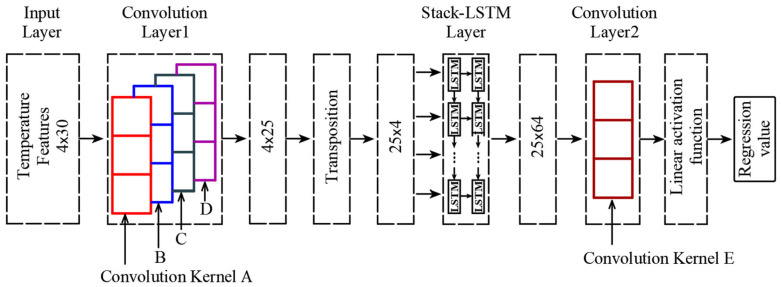
The architecture Stack-LSTM-CNN.

**Figure 3 sensors-25-05346-f003:**
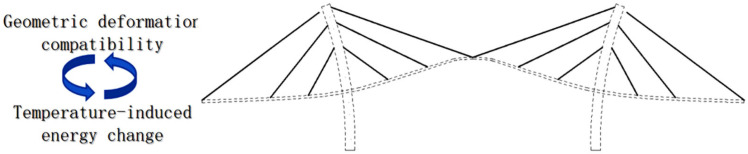
The schematic diagram of the mechanism of the temperature-induced deflection of cable-stayed bridge.

**Figure 4 sensors-25-05346-f004:**
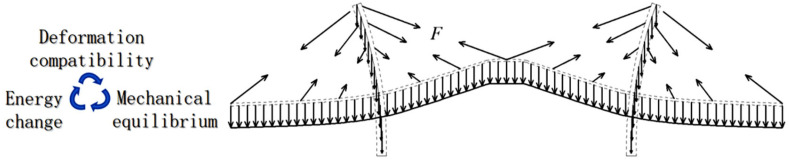
The schematic diagram of the mechanism of the temperature-induced cable tension of cable-stayed bridge.

**Figure 5 sensors-25-05346-f005:**
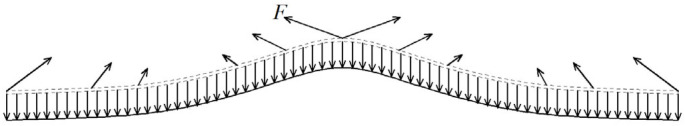
Equilibrium between the cable tension and the gravity of the main girder.

**Figure 6 sensors-25-05346-f006:**
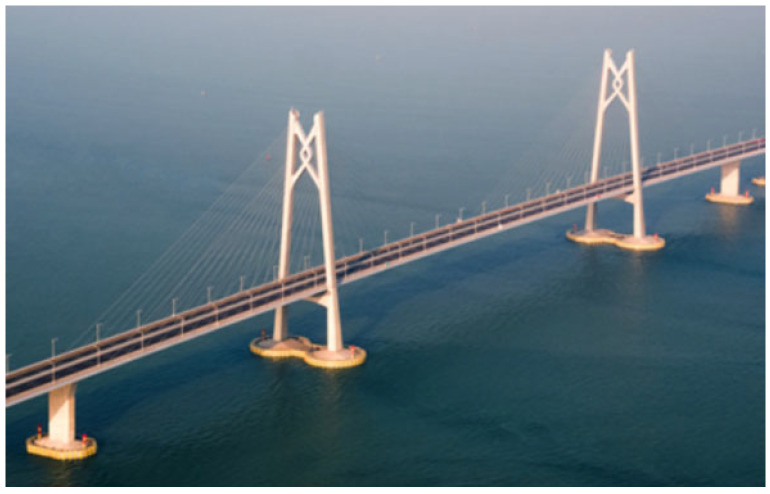
The aerial view of the cable-stayed bridge.

**Figure 7 sensors-25-05346-f007:**
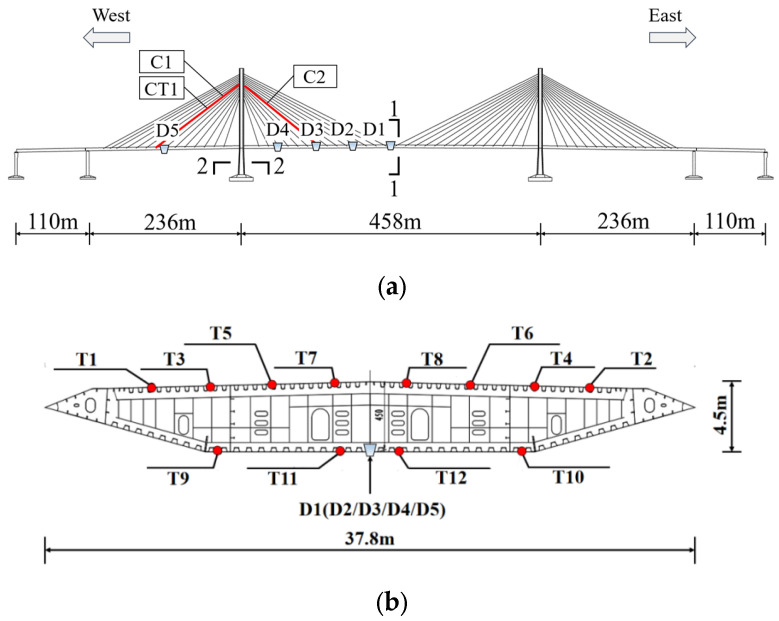
The elevation of the bridge and the section 1-1: (**a**) elevation of the bridge; (**b**) section 1-1.

**Figure 8 sensors-25-05346-f008:**
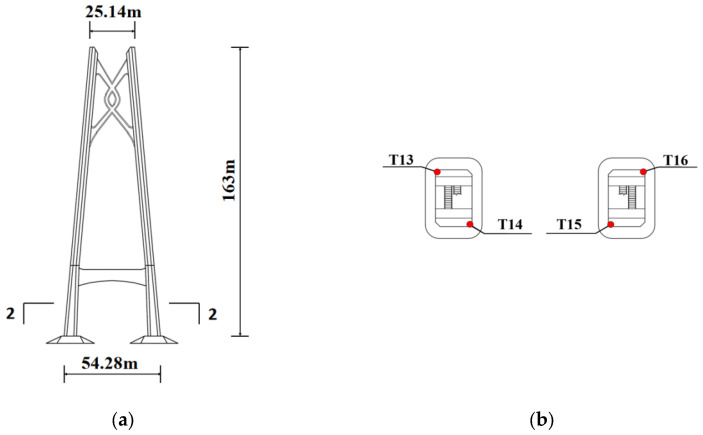
The elevation of the bridge and section 2-2: (**a**) tower elevation; (**b**) section 2-2.

**Figure 9 sensors-25-05346-f009:**
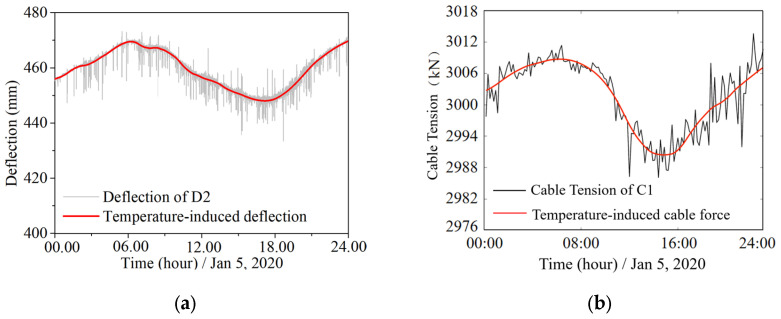
Temperature-induced information in the structural responses: (**a**) the extracted temperature-induced deflection; (**b**) the extracted temperature-induced cable tension.

**Figure 10 sensors-25-05346-f010:**
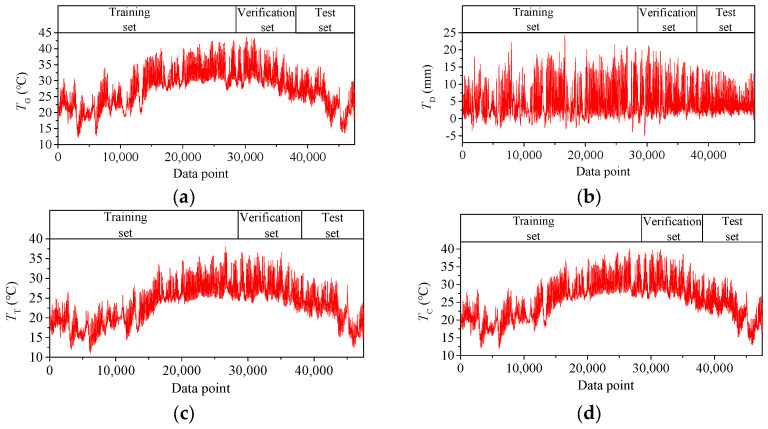
Time–history curves of *T*_G_, *T*_D_, *T*_T_ and *T*_C_: (**a**) the average temperature of the main girder *T*_G_; (**b**) the vertical temperature difference in the main girder *T*_D_; (**c**) the temperature of the tower *T*_T_; (**d**) the temperature of the cable *T*_C_.

**Figure 11 sensors-25-05346-f011:**
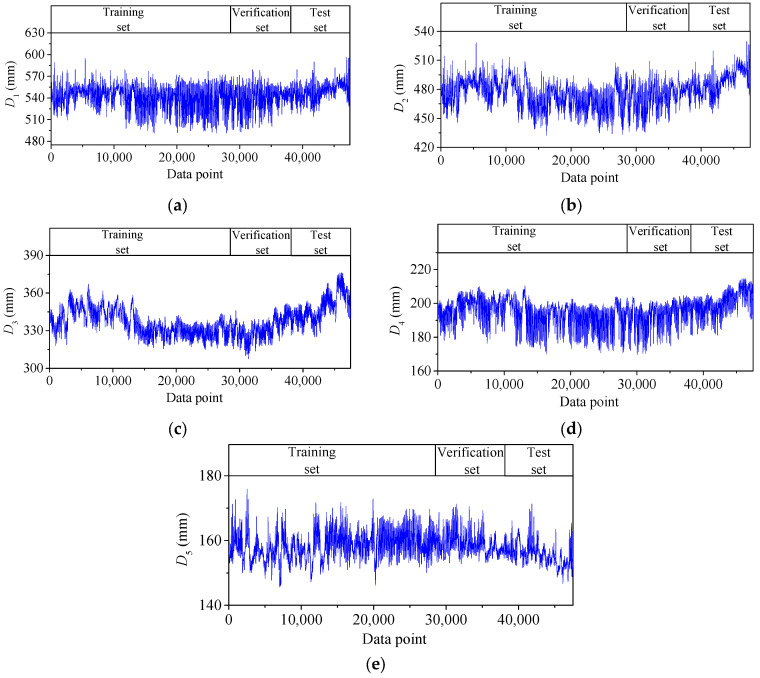
Time–history curves of *D*_1_, *D*_2_, *D*_3_, *D*_4,_ and *D*_5_: (**a**) *D*_1_; (**b**) *D*_2_; (**c**) *D*_3_; (**d**) *D*_4_; (**e**) *D*_5_.

**Figure 12 sensors-25-05346-f012:**
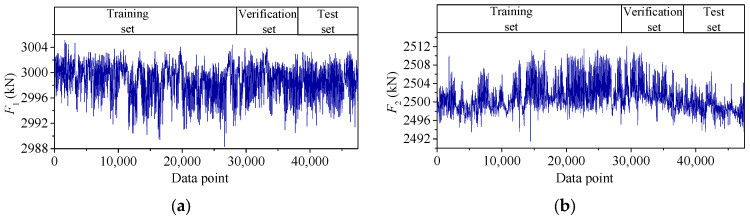
Time–history curves of *F*_1_ and *F*_2_.: (**a**) *F*_1_; (**b**) *F*_2_.

**Figure 13 sensors-25-05346-f013:**
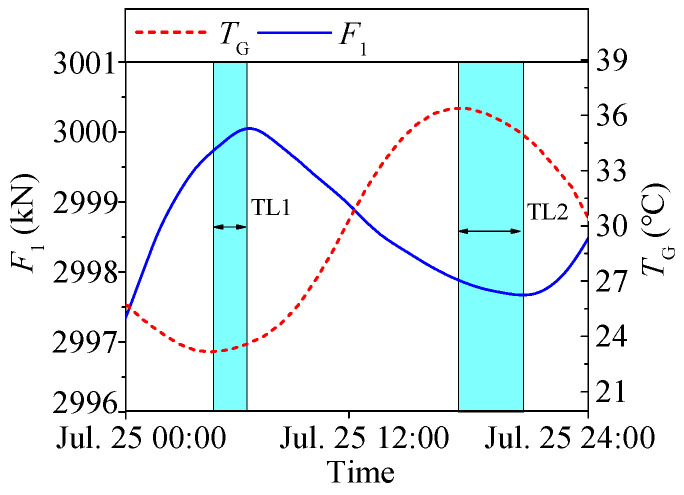
Time lags between temperature-induced tension and temperature feature.

**Figure 14 sensors-25-05346-f014:**
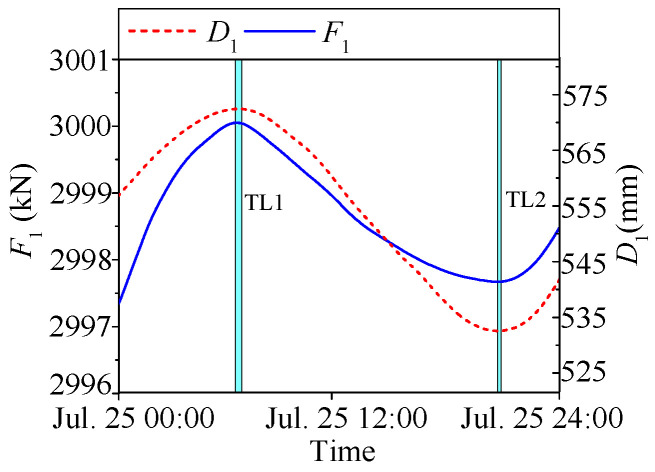
Time lags between temperature-induced deflection and temperature-induced tension.

**Figure 15 sensors-25-05346-f015:**
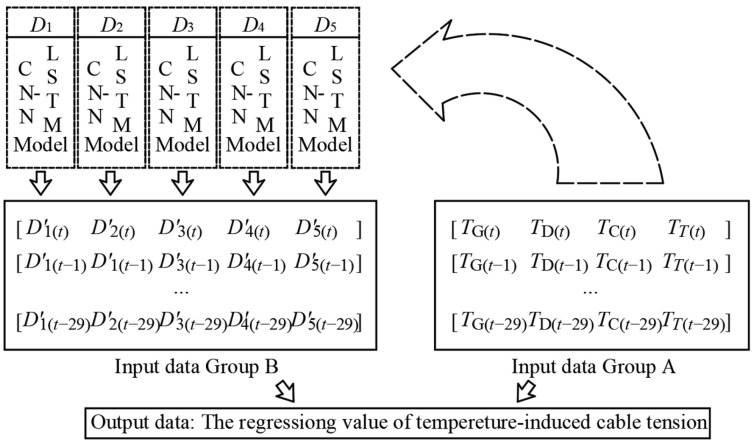
Mixed input data with two data groups composed by regression values of temperature-induced deflection and temperature features.

**Figure 16 sensors-25-05346-f016:**
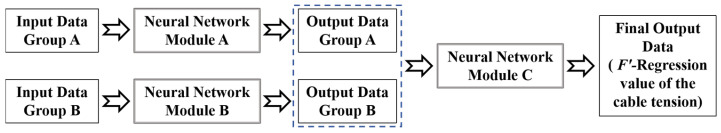
Composite neural networks based on the mixed input data.

**Figure 17 sensors-25-05346-f017:**
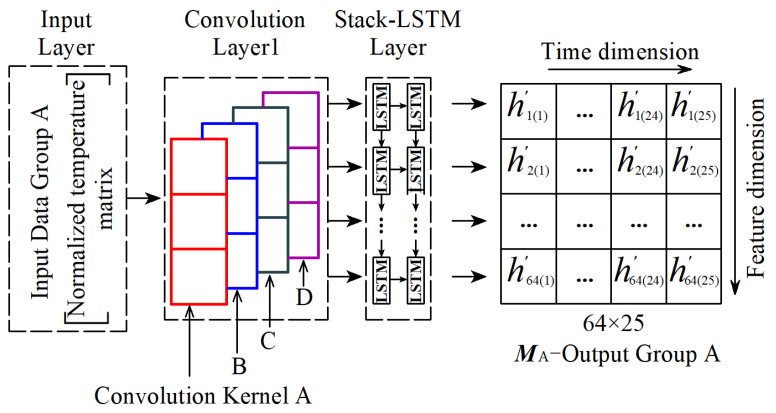
Neural Network Module A and Output Group A by this network.

**Figure 18 sensors-25-05346-f018:**
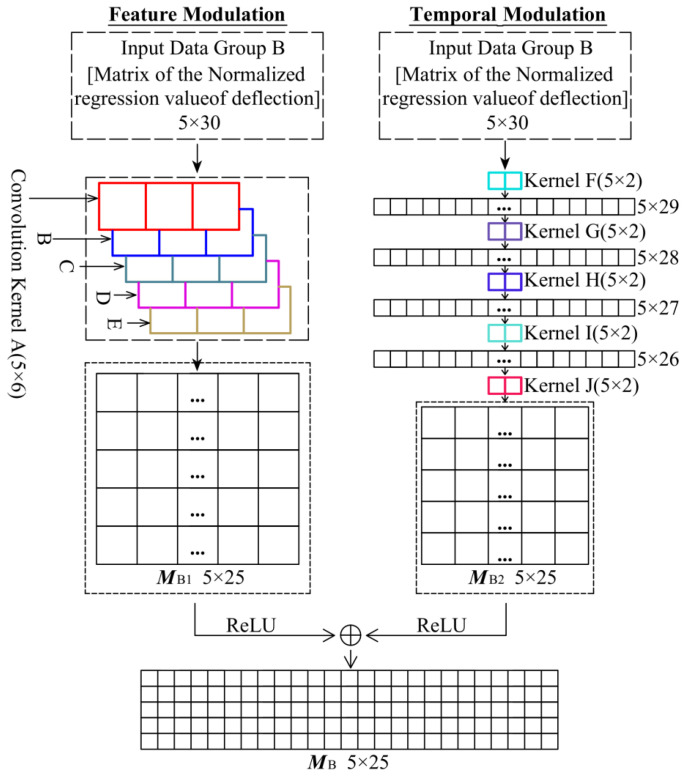
Neural Network Module B and Output Group B by this network.

**Figure 19 sensors-25-05346-f019:**
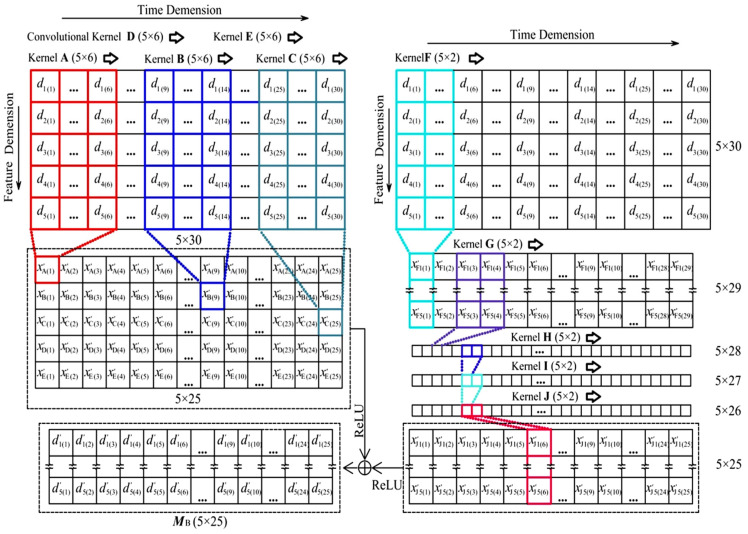
Data flow in the forward propagation of the Neural Network Module B.

**Figure 20 sensors-25-05346-f020:**
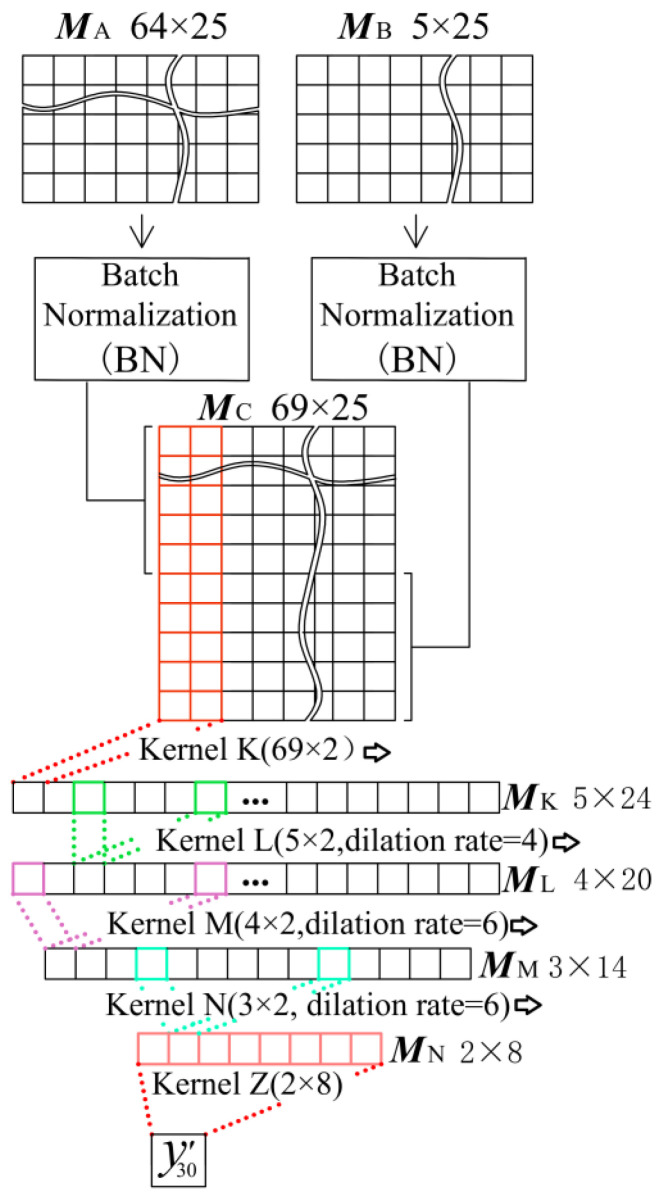
Architecture of Neural Network Module C and the data flow in this network.

**Figure 21 sensors-25-05346-f021:**
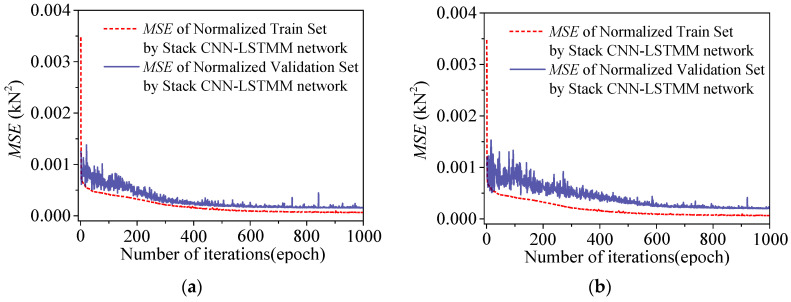
Loss curves in the training process of Stack-LSTM-CNN for the several measurement points of the cable tension: (**a**) loss curve in the training process of Stack-LSTM-CNN of *F*_1_; (**b**) loss curve in the training process of Stack-LSTM-CNN of *F*_2_.

**Figure 22 sensors-25-05346-f022:**
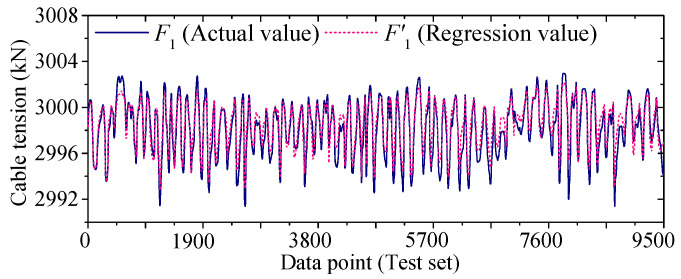
Regression tension *F*’_1_ and actual tension *F*_1_ in test set (by Stack-LSTM-CNN mapping model).

**Figure 23 sensors-25-05346-f023:**
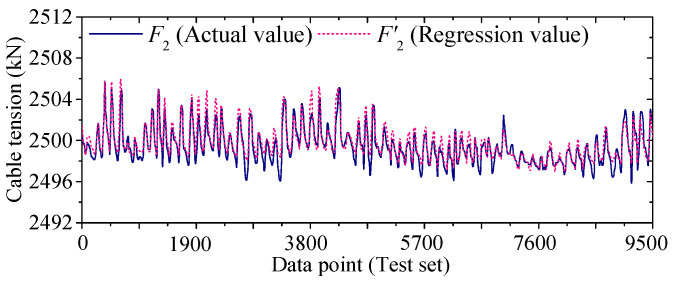
Regression tension *F*’_2_ and actual tension *F*_2_ in test set (by Stack-LSTM-CNN mapping model).

**Figure 24 sensors-25-05346-f024:**
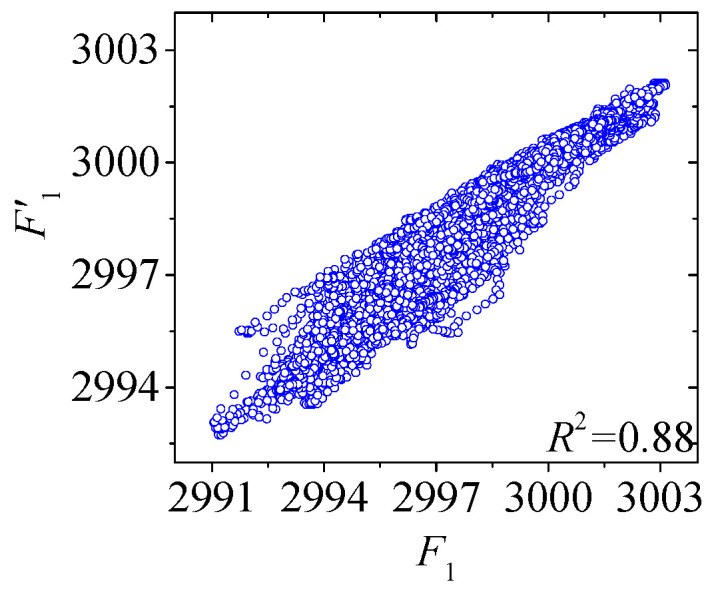
Correlation scatter plot of *F*_1_ and *F*’_1_.

**Figure 25 sensors-25-05346-f025:**
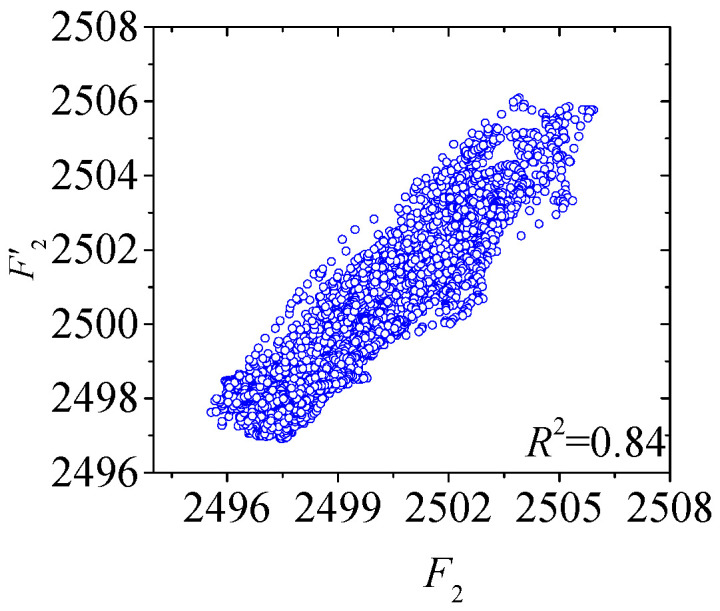
Correlation scatter plot of *F*_2_ and *F*’_2_.

**Figure 26 sensors-25-05346-f026:**
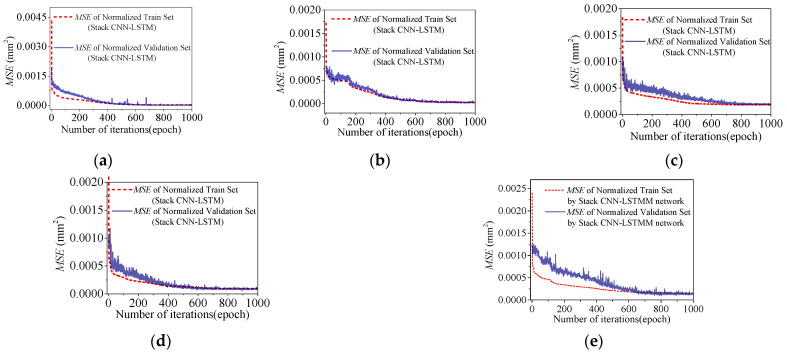
Loss curves in the training process of Stack-LSTM-CNN for the several measurement points of the deflection of the main girder: (**a**) loss curve in the training process of Stack-LSTM-CNN of *D*_1_; (**b**) loss curve in the training process of Stack-LSTM-CNN of *D*_2_; (**c**) loss curve in the training process of Stack-LSTM-CNN of *D*_3_; (**d**) loss curve in the training process of Stack-LSTM-CNN of *D*_4_; (**e**) loss curve in the training process of Stack-LSTM-CNN of *D*_5_.

**Figure 27 sensors-25-05346-f027:**
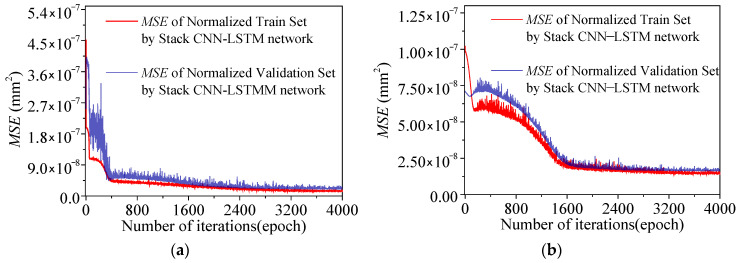
The loss curves of the two models of *F*_1_ and *F*_2_ during the training phase: (**a**) *F*_1_; (**b**) *F*_2_.

**Figure 28 sensors-25-05346-f028:**
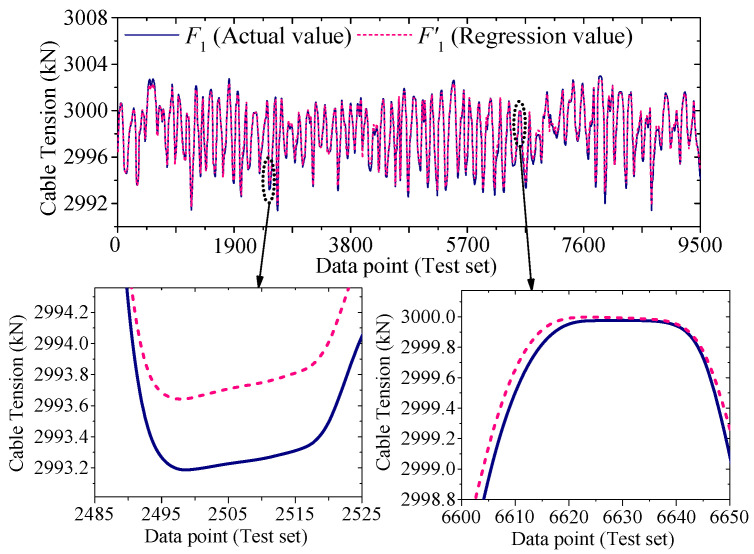
Regression cable tension and actual cable tension of the test set of *F*_1_ (by the mixed input data and the composite neural networks).

**Figure 29 sensors-25-05346-f029:**
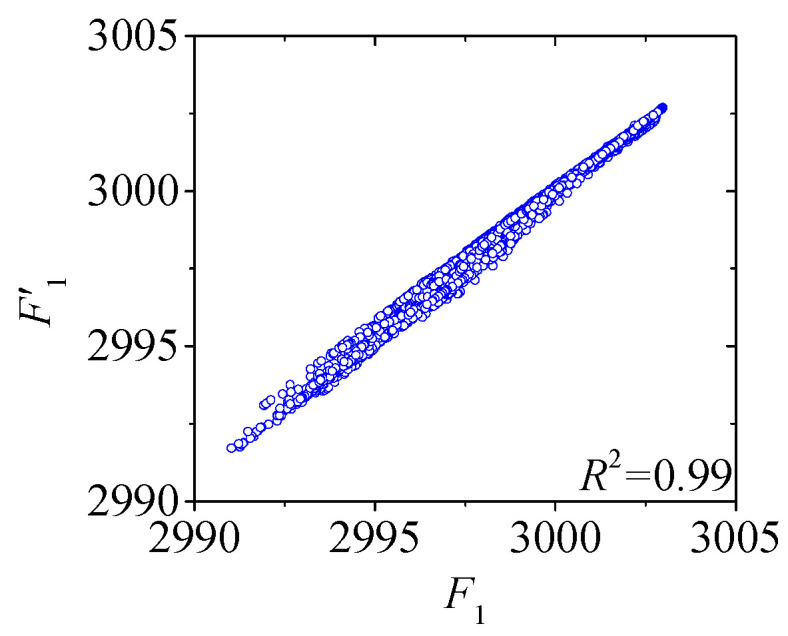
Scatter plot of the correlation between regression cable tension and actual cable tension of the test set of *F*_1_ (by the mixed input data and the composite neural networks).

**Figure 30 sensors-25-05346-f030:**
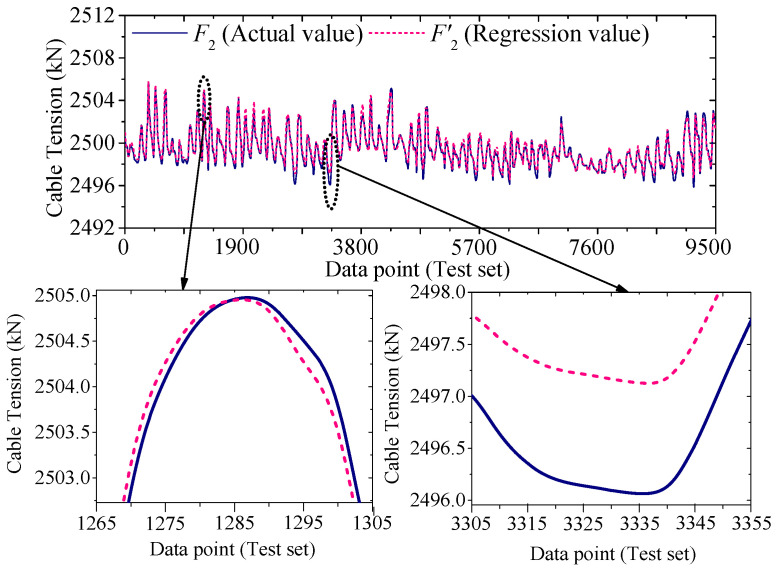
Regression cable tension and actual cable tension of the test set of *F*_2_ (by the mixed input data and the composite neural networks).

**Figure 31 sensors-25-05346-f031:**
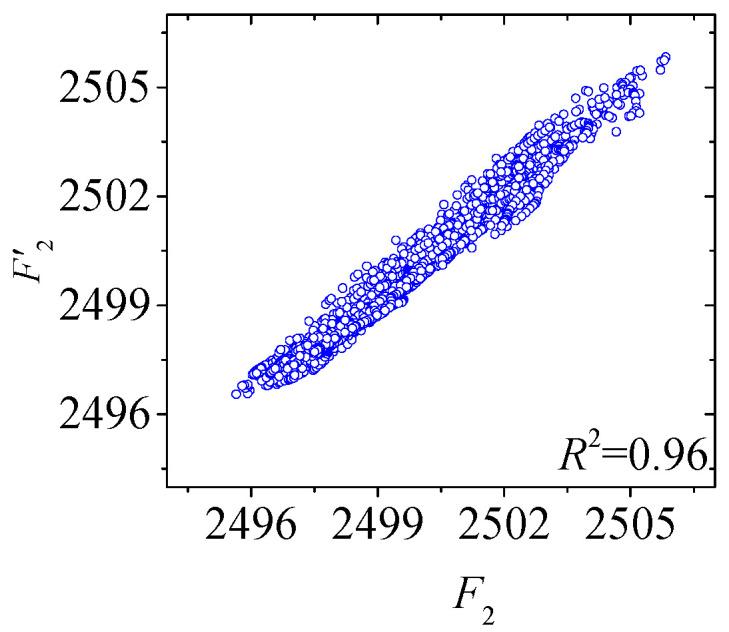
Scatter plot of the correlation between regression cable tension and actual cable tension of the test set of *F*_2_ (by the mixed input data and the composite neural networks).

**Figure 32 sensors-25-05346-f032:**
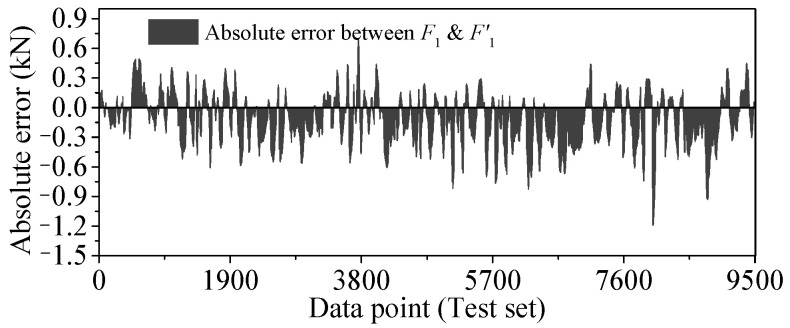
Absolute error (*AE*) of regression value output by Stack-LSTM-CNN mapping model of *F*_1_.

**Figure 33 sensors-25-05346-f033:**
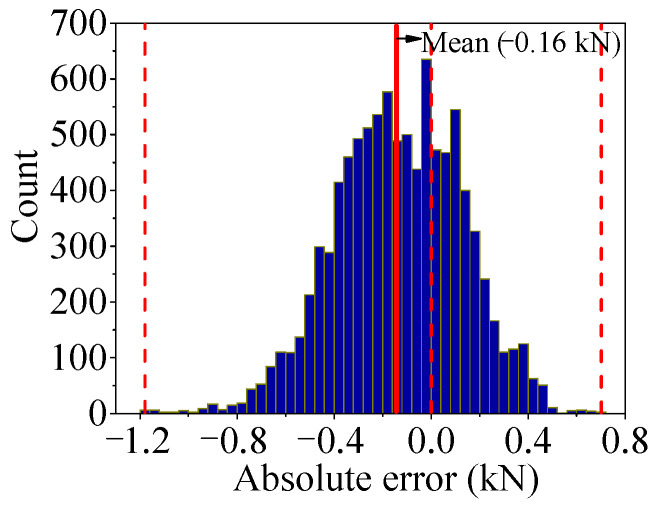
Histogram of *AE* of test set by Stack-LSTM-CNN mapping model of *F*_1_.

**Figure 34 sensors-25-05346-f034:**
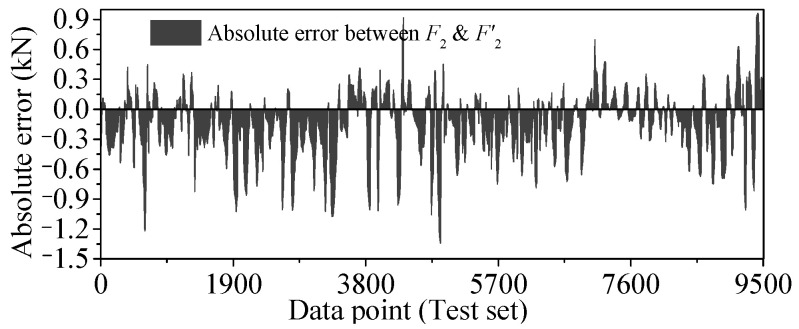
Absolute error (*AE*) of regression value output by Stack-LSTM-CNN mapping model of *F*_2_.

**Figure 35 sensors-25-05346-f035:**
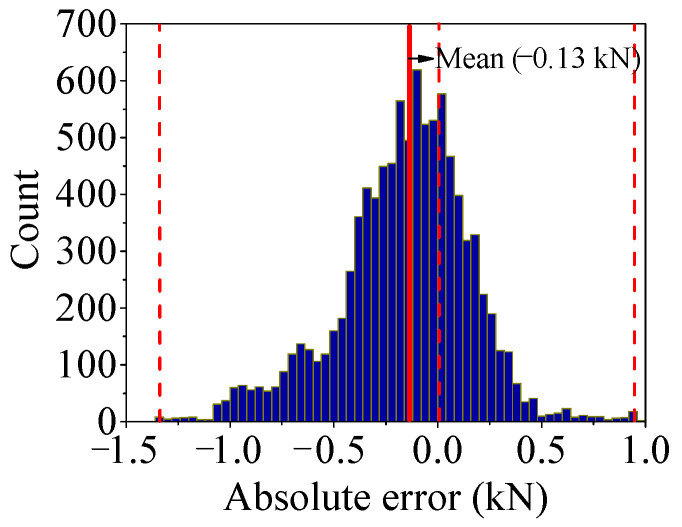
Histogram of *AE* of test set by Stack-LSTM-CNN mapping model of *F*_2_.

**Figure 36 sensors-25-05346-f036:**
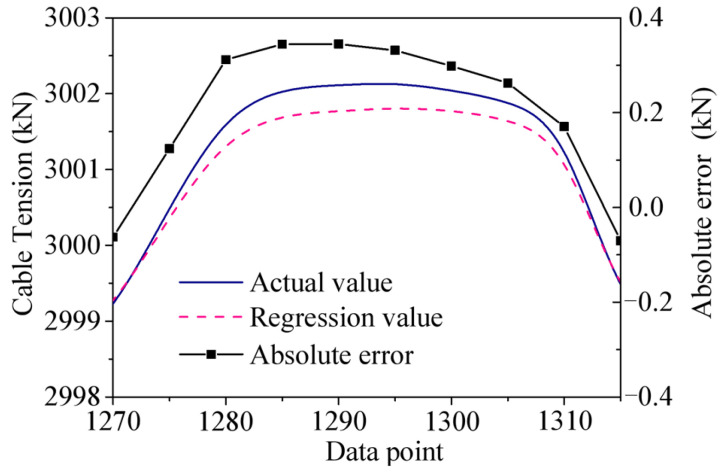
The true value, regression value, and absolute error near the peak.

**Figure 37 sensors-25-05346-f037:**
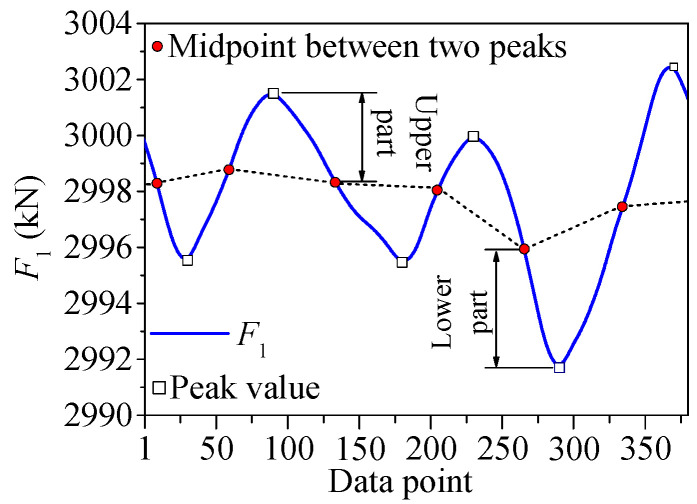
The upper and lower parts of the temperature-induced cable tension.

**Figure 38 sensors-25-05346-f038:**
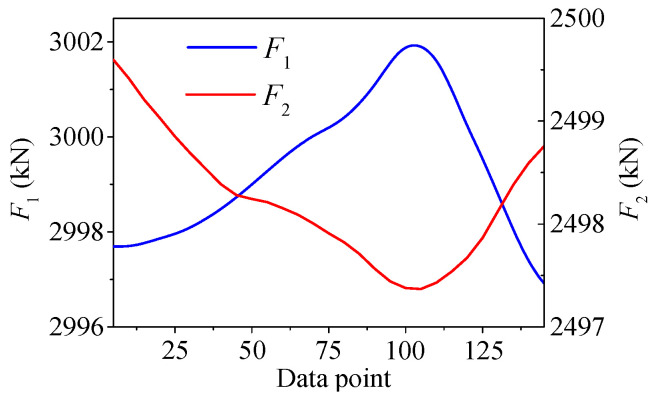
The time-series curves of *F*_1_ and *F*_2_.

**Figure 39 sensors-25-05346-f039:**
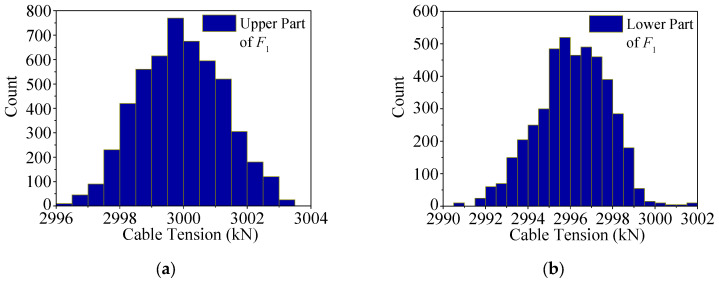
Histograms of the cable tension of the test set of *F*_1_: (**a**) histogram of the cable tension of the upper part of the test set of *F*_1_; (**b**) histogram of the cable tension of the lower part of the test set of *F*_1_.

**Figure 40 sensors-25-05346-f040:**
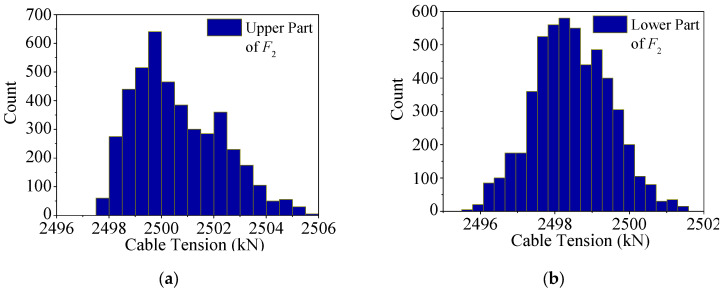
Histograms of the cable tension of the test set of F2: (**a**) histogram of the cable tension of the upper part of the test set of *F*_2_; (**b**) histogram of the cable tension of the lower part of the test set of *F*_2_.

**Table 1 sensors-25-05346-t001:** Comparison of the precision of the Stack-LSTM-CNN models of the different measurement points of the cable tension.

Models	*MSE* (kN^2^)	*MAE* (kN)	*R* ^2^
*F* _1_	1.57 kN^2^	1.01 kN	0.88
*F* _2_	1.12 kN^2^	0.96 kN	0.84

**Table 2 sensors-25-05346-t002:** Comparison of the precision of the Stack-LSTM-CNN models of the different measurement points of the deflection of the main girder.

Models	*MSE* (mm^2^)	*MAE* (mm)	*R* ^2^
*D* _1_	18.86 mm^2^	5.23 mm	0.99
*D* _2_	17.32 mm^2^	4.67 mm	0.98
*D* _3_	16.76 mm^2^	4.55 mm	0.99
*D* _4_	17.45 mm^2^	4.12 mm	0.97
*D* _5_	15.98 mm^2^	4.25 mm	0.99

**Table 3 sensors-25-05346-t003:** Comparison of the precision of the regression models build by the composite neural networks of the different measurement points of the cable tension.

Measurement Point	*MSE* (kN^2^)	*MAE* (kN)	*R* ^2^
*F* _1_	0.086 kN^2^	0.23 kN	0.99
*F* _2_	0.13 kN^2^	0.26 kN	0.96

## Data Availability

Data available on request due to legal.

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
