# Peer review of "Mapping the Complicated Relationship Between a Temperature Field and Cable Tension by Using Composite Deep Networks and Real Data with Additional Geometric Information"

_sensors, 2025, doi:10.3390/s25175346_

Round 1
Reviewer 1 Report
Comments and Suggestions for Authors
This paper investigates the relationship between temperature and cable tension using deep learning methods. While the authors have conducted substantial work, there are several points that warrant further clarification and discussion:
1. The study primarily focuses on neural network methodologies rather than sensor-related aspects. It may be more suitable for publication in AI-focused journals.
2. Have the authors considered the potential thermal effects on the sensors, which could significantly impact the results?
3. Could the authors provide a clearer explanation of the novelty of their approach in comparison to existing neural network methods?
Author Response
We appreciate the diligent work and constructive comments from the editors and reviewers. Please see the attachment to view the detailed responses.

Reviewer 2 Report
Comments and Suggestions for Authors
This paper presents a logical and methodologically sound approach to researching deep learning-based regression models for mapping the relationship between temperature-induced cable tension and the bridge temperature field mixed with temperature-induced responses. Overall, it constitutes a commendable contribution to the field. However, several aspects require further clarification:
-
In Figure 2, the specific input mechanism from the 4×25 matrix to the LSTM layer remains unclear. Please elaborate on this data transformation process, particularly how the dimensional conversion is handled.
-
While the text below Figure 2 indicates normalization of temperature-induced effects, the specific normalization method employed (whether Min-Max standardization, Z-score standardization, etc.) is not specified. This methodological detail is crucial for reproducibility.
-
Figure 6 presents data spanning a 24-hour period but fails to specify the data source (e.g., which specific date, which cable, which beam cross-section deformation and cable force variations are represented). This contextual information is essential for proper interpretation.
-
Wavelet transformation is indeed effective for separating deflection and cable force effects. However, the paper lacks verification that the decomposed deflection and cable force align with actual responses. Evidence through measured data or finite element model validation would strengthen this assertion.
-
The bridge depicted is recognizably the Hong Kong-Zhuhai-Macao Bridge (Qingzhou Channel Bridge), which has a span composition of 165m+458m+165m, totaling approximately 788m. This conflicts with the dimensional data presented in the manuscript. Please explain this discrepancy.
-
The paper employs regression values from a deep learning model of main girder deflection as benchmark values. However, these regression values are essentially predictions whose accuracy requires validation. The manuscript should either provide this validation or reference previous work where such validation has been established.
I look forward to seeing these issues addressed in a revised version of this promising work.
Author Response

(The authors gave the same response as above.)

Reviewer 3 Report
Comments and Suggestions for Authors
This paper proposes a modeling approach for capturing the complex relationship between the temperature field and the cable tension of a cable-stayed bridge using the composite deep networks. This approach provides a novel engineering perspective for fitting the complex data linkage under multi-physics field coupling. Specific comments are as follows:
- The introduction can be appropriately compressed, because the readers in this research field can actually easily understand the engineering significance of the research topic in this paper.
- In Figure 15, for the paragraph “Output data: The regressiong value of tempereture-induced cable tension”, it is recommended to add a text box or underline to this paragraph to visually distinguish it from the figure title below.
- In the section 4.3, it is suggested to compress the text to a certain extent.
- MSE and MAE are commonly used units in error measurement. It is suggested to use language description instead of the equation (7) and equation (8).
5. It is recommended to simplify the text of sections 6.3 and 6.4.
Author Response

(The authors gave the same response as above.)

Reviewer 4 Report
Comments and Suggestions for Authors
This paper approaches the thermal effects on the cable tension in cable-stayed bridges. To this aim, a hybrid model based on artificial neural networks is trained. The subject of the work is very interesting, but, in my opinion, the structural analysis is incomplete.
The main characteristic of a cable-stayed structural system is its non-linear behaviour, due to the second-order effect in the cables. However, although the work puts emphasis on the geometric information, the geometrically non-linear behaviour of the cables is not considered at all.
For a temperature-induced cable tension analysis as approached in this work, more even in the context of a physically-informed machine learning strategy, the geometrically non-linear structural analysis plays a key role, since it allows to predict the real stiffness of the cable and, consequently, the resulting stress tensor under service loads. Moreover, these cables are usually prestressed, which also affects the second order structural response. Thus, for example, in the case of slender cable-stayed steel footbridges, where their dynamic behaviour is very relevant from the point of view of the pedestrian comfort, this type of analysis allows to determine the modification of the dynamic response of the structure due to the change in the stress level of the cables.
Due to the complexity of such analysis, a full non-linear finite element model of the case-study presented in the Section 4.1 (including the deck, the towers and the cables that connect them) may be developed. In order to simplify the model, the deck may be approached using beam elements.
Once the finite element model has been performed, this may be updated through, for example, a dynamic modal analysis. This updated model allows to generate a more efficient input features database for the training of a regressor regarding to the second order behaviour of the cables (and considering thus the full displacement path of the complete structural system).
As conclusión, I encourage the authors to review and complete the paper, with a comprehensive analysis of all the physical concepts involved in the problem under consideration, in the terms indicated above.
Author Response

(The authors gave the same response as above.)

Round 2
Reviewer 1 Report
Comments and Suggestions for Authors
I have no more questions.
Reviewer 2 Report
Comments and Suggestions for Authors
All the questions I raised were treated seriously and answered in detail. The supplementary information provided was reasonable and comprehensive. Thank you!